# Pharmacological Approaches to Controlling Cardiometabolic Risk in Women with PCOS

**DOI:** 10.3390/ijms21249554

**Published:** 2020-12-15

**Authors:** Andrea Cignarella, Roberto Mioni, Chiara Sabbadin, Francesca Dassie, Matteo Parolin, Roberto Vettor, Mattia Barbot, Carla Scaroni

**Affiliations:** 1Department of Medicine, University of Padova, 35128 Padova, Italy; robertomioni@libero.it (R.M.); chiarasabbadin.85@gmail.com (C.S.); francesca.dassie@libero.it (F.D.); roberto.vettor@unipd.it (R.V.); mattiabarbot85@gmail.com (M.B.); carla.scaroni@unipd.it (C.S.); 2Clinica Medica 3, University Hospital, 35128 Padova, Italy; 3Endocrinology Unit, University Hospital, 35128 Padova, Italy; 4Internal Medicine 2, Ca’ Foncello Hospital, ULSS2 Marca Trevigiana, 31100 Treviso, Italy; matteoparolin@hotmail.it

**Keywords:** biomarkers, anti-androgens, hypoglycemic agents, combined oral contraceptives, statins

## Abstract

Polycystic ovary syndrome (PCOS) is characterized by elevated androgen production and subclinical changes in cardiovascular and metabolic risk markers. Total cholesterol, high-density lipoprotein (HDL) cholesterol, fasting glucose, and fasting insulin appear to increase specifically in PCOS compared with fertile women. PCOS also confers an increased risk of cardiometabolic disease in later life. Novel biomarkers such as serum’s cholesterol efflux capacity and blood-derived macrophage activation profile may assist in more accurately defining the cardiometabolic risk profile in these women. Aldosterone antagonists, androgen receptor antagonists, 5α-reductase inhibitors, and synthetic progestogens are used to reduce hyperandrogenism. Because increased insulin secretion enhances ovarian androgen production, short-term treatment with metformin and other hypoglycemic agents results in significant weight loss, favorable metabolic changes, and testosterone reduction. The naturally occurring inositols display insulin-sensitizing effects and may be also used in this context because of their safety profile. Combined oral contraceptives represent the drug of choice for correction of androgen-related symptoms. Overall, PCOS management remains focused on specific targets including assessment and treatment of cardiometabolic risk, according to disease phenotypes. While new options are adding to established therapeutic approaches, a sometimes difficult balance between efficacy and safety of available medications has to be found in individual women.

## 1. Introduction

Polycystic ovary syndrome (PCOS) is one of the most common endocrine disorders affecting women in their reproductive age. PCOS is a clinical heterogeneous condition defined by oligo- or anovulation, elevated androgen production, and polycystic ovary changes as per the Rotterdam criteria [1]. Apart from classic symptoms, including acne, hirsutism, and infertility, PCOS carries important cardiovascular comorbidities such as insulin resistance (IR), dyslipidemia, obesity, and hypertension [2,3]. Despite the lack of solid longitudinal studies, epidemiological data confirmed an increased prevalence of cardiovascular (CV) events in post-menopausal women with a clinical history of irregular menstrual cycles [4]. It is therefore paramount to promptly recognize and revert subtle cardiometabolic impairment to reduce future CV events. Although PCOS is a multifactorial disease with unknown pathophysiology, IR and compensatory hyperinsulinemia play a central role in the development and maintenance of both gynecological and metabolic disorders [5]. IR is indeed a common finding in PCOS, even though it is not found among diagnostic criteria, and represents a well-known predictor for future development of type 2 diabetes (T2DM). One of the main causes of IR is overweight and obesity, whose prevalence can reach 80% in PCOS, but even lean patients often display relative hyperinsulinemia with consequent increased likelihood of further developing T2DM [6]. Besides glucose metabolism impairment, hyperinsulinemia favors other classical CV risk factors such as hypertension, resulting from vascular smooth muscle cell hypertrophy and endothelial dysfunction through endothelin-1, and dyslipidemia [7]. Insulin has a gonadotropin-like action and may increase androgen production, which contributes to worsening both metabolic impairment and clinical picture [8]. In this regard, IR is an appealing target for PCOS management, and several insulin-sensitizing drugs have been used to treat patients with PCOS. However, it should be recalled that lifestyle changes should be the basis of any treatment because, if effective, they could significantly reduce cardiometabolic risk in later life [9].

The pathophysiology of PCOS is complex and several treatment options are currently available. In this comprehensive review, we set out to discuss the integration of novel biomarkers with pharmacological intervention to improve PCOS management. We summarize current treatment strategies, including discussion about disagreements in the field, and suggest future clinical trajectories.

## 2. Data Collection and Methodology

A comprehensive search of the literature was conducted by using the PubMed database by all authors. The search strategy included terms related to PCOS, cardiometabolic risk factors, and pharmacological treatments. Clinical experience of the authors of sections dealing with pharmacological treatments also guided content selection.

## 3. Emerging Cardiometabolic Risk Biomarkers

Elucidating the factors that increase CV risk in PCOS would help refine current therapeutic strategies. Traditional CV risk factors have received attention [3] but are associated with emerging sex-specific risk factors such as adverse pregnancy outcomes (e.g., pre-eclampsia and gestational diabetes), ovarian failure, and risk factors outside the realm of pregnancy and menopause such as PCOS, breast cancer, autoimmune disease, and hormone-based contraceptive methods [10]. Based on a recent metanalysis, total cholesterol, high-density lipoprotein (HDL) cholesterol, fasting glucose, and fasting insulin levels appear to increase specifically in women with PCOS compared with fertile women [11]. It has been suggested that low-grade inflammation in PCOS stimulates polycystic ovaries to enhance their androgen production and secretion, and this is one reason for elevated risk of heart and blood vessel problems. However, cardiometabolic risk in PCOS women is multifactorial and could in some cases also include gene defects [12]. Therefore, the assessment of additional CV risk biomarkers may assist in identifying the optimal treatment strategy for such patients and may also result in a paradigm shift by providing potential novel therapeutic targets. 

Both proinflammatory activation and dysregulated cholesterol metabolism in the monocyte–macrophage system are key players in vascular inflammation and atherosclerosis [13]. Estrogenic pathways and menopausal status are important determinants of human blood-derived macrophage activation [14,15]. The ability of HDL particles to promote macrophage cholesterol efflux is considered the main cardioprotective function associated with HDL. Therefore, stimulation of the ATP-binding cassette transporter-A1 (ABCA1)-dependent cholesterol efflux from macrophage cells to HDL, rather than HDL cholesterol (HDL-C) levels per se, was found to be inversely associated with the incidence of atherothrombotic CV events [16,17]. Monocyte–macrophage activation was investigated in women diagnosed with PCOS, and this profile was compared with that in women with regular menstrual cycles at follicular (F) and luteal (L) phases. In addition, the capacity of apoB-100-depleted sera to induce cholesterol efflux from macrophage cell models was assessed and compared between healthy women and women with PCOS. Previous reports suggested attenuated cholesterol efflux capacity (CEC) that was combined with atherogenic lipoprotein distribution in women displaying PCOS [18,19]. These findings were refined and expanded by analyzing specific pathways for cholesterol egress from macrophages as well as analyzing in more detail macrophage function and effects of menstrual cycle [20]. Experimental procedures to explore novel biomarkers include flow cytometry for M1/M2 macrophage identification as well cell culture methods for cholesterol efflux via aqueous diffusion, ABCA1, and ABCG1 pathways. These approaches are relevant and have been validated. Of note, macrophage responses to lipopolysaccharide (LPS) and cytokines in PCOS were significantly attenuated when compared with women having normal menstrual cycle. There was a significant elevation in the classically activated macrophage subfraction in response to LPS and interferon-γ (M1) stimuli in both the F- and L-phase that was not observed among PCOS women. In addition, cholesterol efflux to apoB-depleted serum from PCOS women via multiple routes was significantly impaired compared to that of healthy women. Evidence for HDL dysfunction is not apparent as HDL CEC impairment was related to serum HDL concentration. However, when CEC was adjusted for HDL-C levels, the difference was lost regarding ABCA1- and ABCG1-CEC, but this difference remained for total CEC and aqueous diffusion (Figure 1). The latter mechanism is reportedly one of the major contributors to cholesterol efflux and is a dominant mechanism for efflux from macrophages containing normal cholesterol levels. As HDL-C levels were significantly lower in PCOS women having elevated testosterone levels, there might also be differences in HDL subpopulation distribution due to an androgen effect that would not favor cholesterol efflux [21]. This aspect should be explored in future studies.

Investigating the effects (if any) of prescribed medications on these biomarkers in PCOS women would yield a more accurate assessment of cardiometabolic risk including aspects of cell activation at the vascular immune interface [22].

## 4. Antiandrogen Therapy

Although the mechanism by which androgens, in particular testosterone or dihydrotestosterone (DHT), influence the CV system and metabolism in women is still debated and not completely understood, an increasing number of studies show a pathophysiologic link between androgen signaling and vascular and metabolic disorders, such as hypertension, ischemic stroke, aortic aneurysm, peripheral artery disease, atherosclerosis, altered glycemic control, and lipid metabolism in PCOS patients [23,24,25]. The hyperandrogenic syndrome is one of the most common endocrine diseases, affecting up to 18% of women during their reproductive years. 

PCOS not only has a negative effect on reproductive potential, but it is also characterized by high prevalence of obesity (especially visceral obesity), which ranges from 38 to 87% and is considered an expression of both ovarian hyperandrogenism and IR [26]. As the tight link between obesity, IR/hyperinsulinism, and increased cardiometabolic risk is well known, the role of hyperandrogenism in dyslipidemia, diabetes, and CV disease (CVD) remains difficult to interpret [27]. In fact, guidelines from national and international societies recommend lifestyle changes (diet and physical activity) as a first step to reduce weight, when in excess, or to improve dyslipidemia or diabetes in women with PCOS or androgen excess [27,28]. Weight loss frequently results in improved metabolic parameters, blood pressure, and hyperandrogenism [28,29]. Based on its high prevalence in the general population, and the correlation with other features of CVD and metabolic syndrome, PCOS is a good model to investigate the impact of androgen excess on cardiometabolic risk. However, the improvement or correction of hyperandrogenism has always been considered as a second-line therapeutic goal with respect to fertility or obesity, even though in real clinical practice the psychological consequences of dermatological signs associated with higher androgen levels are recognized [30,31]. In women with androgen excess or in PCOS subjects, specific antiandrogenic or 5-alpha-reductase inhibitor drugs do not usually represent a first-line therapeutic option, and only few studies have correlated the use of these drugs with improvement of androgen-related cardiometabolic parameters.

The treatment course for PCOS women largely depends on the severity of symptoms and the specific therapeutic goals. Treatment against hirsutism, acne, or androgenic baldness frequently occurring in PCOS usually aims at decreasing androgen excess and ameliorating these symptoms. However, long-term consequences, such as the risk of increasing weight or developing metabolic and cardiovascular abnormalities, must be considered. Estrogen-progestin combination therapy remains the predominant treatment for the reduction of hyperandrogenism and will be discussed in Section 7.

This section, dealing with antiandrogen drug effect on cardiometabolic disease in PCOS women, aims to bring out a more complex picture of the management of hyperandrogenism in women. Long-term research in this area that is needed for the development of more consistent models and controlled experimental designs that will provide insights into the impact of endogenous androgen concentrations and the appropriate dosing of antiandrogen drugs as an effective strategy against androgen excess and its deleterious effect on cardiometabolic risk in women.

### 4.1. Flutamide

Flutamide (FLU) is a non-steroidal androgen receptor (AR) antagonist. It is metabolized to hydroxyflutamide, which has a half-life of 6–8 h and mediates the antiandrogenic effects [32]. The drug acts mainly at peripheral level blocking competitively the cytoplasmatic and nuclear binding of androgens to the AR. The pure antiandrogenic effects of FLU are superior to those of spironolactone or cyproterone [33]. However, FLU might also reduce androgen synthesis. The recommended dosage is 62.5 to 750 mg daily in combination or not with oral contraceptives [34]. Guidelines and a position statement support FLU use for treatment of hirsutism, but it is considered an off-label drug for the treatment of female hyperandrogenism and always requires the patient’s informed consent [35]. Although FLU has demonstrated benefit in the management of hirsutism, particularly when used in combination with oral contraceptives, its use is limited by the risk for significant hepatotoxicity and teratogenicity [35,36]. However, more recent reviews indicate that FLU, when given at low dose (~1 mg/kg daily), can be considered a first-choice balance between antiandrogenic efficacy and hepatic safety for women and adolescent girls, especially those affected by PCOS, with hirsutism or other androgen-related symptoms [37,38]. The Androgen Excess (AE) and PCOS Society recommends daily doses of less than 250 mg, but several authors confirm a beneficial and safe effect on hirsutism and acne in PCOS patients at doses between 62.5 and 125 mg/day, even for prolonged intervals beyond 12 months [38,39,40]. 

In monotherapy, FLU has cosmetic benefits; improves the effect of hyperandrogenism; and induces, as secondary effects, a reduction of serum LDL-C cholesterol and an increase in HDL-C, even in non-obese adolescents. Hyperandrogenic PCOS women are characterized by upper body obesity independently of weight, and exogenous androgen leads to increased visceral fat accumulation and decreased serum HDL via a direct modulation of lipoprotein lipase and lipolysis. It has been demonstrated that FLU, at the dose of 500 mg/day for 4 months, decreases total cholesterol (TC) (−18%), LDL-C (−13%), LDL/HDL ratio (−23%), and triglyceride (TG) levels (−23%), suggesting that this antiandrogenic drug improves lipid profile via direct inhibition of androgenic actions, independently from obesity [41,42]. In addition, androgens worsen insulin resistance in women. Accordingly, FLU, even at medium–low doses (125–250 mg/day), was able to improve not only hirsutism or acne, but also hyperinsulinemic state and IR in PCOS subjects. On the other hand, several authors showed that FLU was not very effective in improving IR, when administered alone, while if associated with metformin, a well-known insulin-sensitizing drug, FLU significantly improves both IR indices and visceral adipose distribution, mobilizing visceral fat at abdominal level, and finally increasing lean mass [43,44]. This mobilization of visceral adipose tissue represents one of the mechanisms by which FLU may exert a modulating effect on blood pressure in hyperandrogenic PCOS subjects. Considerable evidence shows that androgens play a pivotal role in gender-associated differences in blood pressure (BP) regulation, and several experimental models showed that androgens can promote hypertension [45,46]. FLU has been shown to lower blood pressure in PCOS experimental models [47,48]. Ayossa et al. [49] disclosed that FLU decreases and normalizes vascular resistance in the uterine artery, suggesting that this compound both reduces BP and regulates blood flow to the reproductive organs in PCOS patients.

### 4.2. Bicalutamide

Bicalutamide (BCL) is a non-steroidal antiandrogen that binds to AR in androgen-sensitive cells and prevents the physiological effects of DHT. Its long elimination half-life (t_1⁄2_) permits once-daily administration. In most, although not all, in vitro studies BCL had a 2- to 4-fold higher affinity for AR than hydroxyflutamide. It is currently commercially available in Europe, and it has been tested for male patients with advanced prostatic carcinoma. Based on clinical trials for advanced prostate cancer and benign prostatic hyperplasia, hepatic adverse events occurred in 1.7% of patients and no cases of hepatic failure were reported. Other events reported during BCL therapy include breast discomfort, diarrhea, vomiting, nausea, hematuria, asthenia, and skin changes such as rash. As these side effects rarely occur, BCL can considered an AR antagonist with a better safety and tolerability profile compared with FLU [50,51]. BCL is a relatively new and still poorly investigated drug for the management of women’s hyperandrogenism. 

Although BCL is a potent antiandrogen drug, with excellent safety and stability of plasma and tissue concentration, the “protective” effects against androgen-dependent CV risk in women with PCOS are less evident. Recent data showed that treatment with 25 mg/day BCL for at least 6 months did not induce any changes in fasting serum glucose, fasting insulin, HOMA-IR, prolactin, and C-reactive protein (CRP) levels in young hyperandrogenic women with PCOS [52]. In the latest randomized, double-blind, controlled study testing the effect of 50 mg/day BCL associated with COC, Moretti et al. [53] have demonstrated the efficacy and tolerability of BCL in the treatment of severe hirsutism in young PCOS patients after 18 months, but without observing any changes in blood pressure, fasting insulin, glucose, waist circumference, and body composition. TC, HDL-C, LDL-C, and TG were also considered in the same study, all of which were found to be elevated. This can be justified by the simultaneous use of COC. BCL use in PCOS women is off-label and further studies are required to assess its potential role in controlling cardiometabolic risk.

### 4.3. Spironolactone

Spironolactone (SPA) is one of the most commonly used antiandrogen agents in the United States and in Europe [54] and works primarily by antagonizing the effects of aldosterone at renal mineralocorticoid receptors (MR). It also competes with DHT for binding to AR, although its effect on the latter is minimal compared to DHT. SPA also acts directly through blocking 5α-reductase-activity, competing with androgens for binding to SHBG, blocking conversion of testosterone to DHT in dermal papilla cells, and antagonizing the androgenic effect of DHT on the hair follicle [55]. SPA seems to also reduce levels of both GnRH and LH, thereby attenuating LH-stimulated androgen secretion. SPA is a relatively effective treatment, in particular for acne and hirsutism. Side effects such as hyperkaliemia, headaches, breast tenderness, dry skin, and gastritis have been reported. However, the most frequent side effect is intermenstrual spotting, which can lead to treatment discontinuation. Because SPA, as other antiandrogen drugs, has the potential for teratogenicity (inadequate masculinization of male genitalia), contraceptive therapy is highly recommended. The dosage of SPA is usually 100–200 mg daily, given in two divided doses [56,57,58]. 

Endothelial dysfunction, the initial step in the process of atherogenesis, has been reported to occur in young PCOS patients, consistent with the notion that ARs are expressed in the vessel wall and testosterone is known to worsen endothelial function in females. These hyperandrogenic subjects are also at increased risk for CVD. SPA treatment for 6 to 12 months has been reported to improve the lipid profile and reverse endothelial dysfunction in PCOS patients [59]. However, this result must be considered with caution. In fact, aldosterone can also impair endothelial function through increased NADPH oxidase activity and mitochondrial generation of reactive oxygen species, which decrease bioavailability of nitric oxide. As SPA can antagonize aldosterone by blocking its receptor, it cannot be excluded that the improvement on endothelial function exerted by SPA occurs via an anti-aldosterone mechanism. In PCOS women, hypertension alters vascular smooth muscle cells causing vascular muscle wall hypertrophy with reduced compliance and interference with endothelium-dependent vasodilatation mechanisms. Increased arterial stiffness, a parameter independent of age and systolic blood pressure, implies a possible direct effect of PCOS components on the vascular wall [60,61]. SPA treatment at 50–100 mg was reported to ameliorate vascular compliance in both pre- and post-menopausal PCOS subjects. In particular, SPA improved large artery elasticity and induced a highly significant reduction in systemic vascular resistance, without changes in small artery elasticity [62]. However, it remains to be clarified whether this effect of SPA on vascular compliance is mediated by ARs or MRs, which are both expressed in the endothelial and smooth muscle cells of arterial vessels.

### 4.4. Ketoconazole

Ketoconazole (KTZ) is an orally active antifungal agent of the imidazole class. In vivo and in vitro studies showed that this agent acts as a potent inhibitor of both gonadal and adrenal steroidogenesis [63]. In humans, KTZ inhibits the P450 enzymes in the adrenal gland involved in cortisol production, which led to EMA approval in the management of Cushing’s syndrome [64]. Cedeno et al. [65] showed that KTZ treatment for 10 days at 400 mg or 800 mg/day in 10 PCOS patients led to a significant improvement in acne and hirsutism, as well as a significant reduction of cholesterol (TC, LDL-C, and LDL/HDL ratio) and lipoproteins levels. In detail, 400 mg/day of KTZ therapy reduced TC by about 10%, LDL-C by 13%, and increased apo-AI by about 7%. At higher dose (800 mg/day), KTZ lowered the LDL/HDL ratio by 40%, and reduced LDL-C by 33% and apo B by 21%. However, the KTZ-induced lipid-lowering effect may be secondary to increased estrogen levels through direct stimulation of aromatase activity. KTZ does not appear to exert a significant effect on glycemic control. Other authors have reported modified biochemical liver parameters, thus confirming that KTZ should be used off-label with caution in PCOS women [66]. In addition, KTZ causes reversible non-competitive inhibition of cytochrome P450 (CYP) 3A4, which greatly increases the risk of interaction when other drugs that are substrates of this enzyme are taken concomitantly.

### 4.5. Finasteride

Finasteride (FND) is a competitive inhibitor of steroid 5-alpha-reductase 2, an enzyme that catalyzes the conversion of testosterone to DHT. FND neither directly inhibits testosterone synthesis nor interacts with AR. 5α-reductase inhibitors increase circulating levels of testosterone and its conversion to estradiol [67]. While improving hair thickness and growth in men affected by baldness, even at low dose (1 mg/day), FND has not been consistently successful in treating normoandrogenic female patients with androgenic alopecia, suggesting that the therapeutic effect of FND is observed only at higher androgen levels [68]. A large body of evidence suggests that FND improves hirsutism scores in hyperandrogenic women affected by PCOS with hirsutism, acne, and androgenic alopecia [69,70]. To date, no studies have assessed the role of FND in controlling CV risk factors in humans, despite preclinical evidence of benefit in rodent models [71,72]. In a group of PCOS women, Lackryc et al. [70] found no changes in blood pressure and heart rate after 6 months of FND treatment (5 mg/day). Similarly, no changes in metabolic parameters such as carbohydrates and/or lipids after FND treatment in hyperandrogenic women have been reported [56,69,73]. By contrast, recent data have shown that FND (5 mg/day) improves IR (HOMA-IR) as well as glycemic and insulinemic responses after OGTT (AUC-glucose and AUC-insulin until 180 min) independently of weight reduction, thus suggesting FND as an alternative therapy for PCOS-related metabolic disease [74].

## 5. Antidiabetic Medications

### 5.1. Metformin

Among antidiabetic drugs, metformin is the most widely used in PCOS, due to its safety and tolerability. Despite this, its prescription is still off-label and the Endocrine Society Guideline recommended its use in PCOS women with T2DM or impaired glucose tolerance (IGT) only when lifestyle changes prove unsuccessful [75]. 

Metformin is a member of biguanide family. It exerts its anti-hyperglycemic effects through several mechanisms: it increases cellular insulin sensitivity and peripheral glucose uptake by upregulating GLUT-4, decreases hepatic gluconeogenesis, and reduces the amount of glucose reabsorption from the bowel [76]. 

Metformin is the first-line treatment in T2DM because it acts at the hepatic level reducing IR, which is extremely frequent in PCOS too. Although not as effective as lifestyle modification, metformin is able to reduce the progression to overt T2DM in PCOS. Despite having virtually no effect on BMI, metformin was associated with mild weight loss in some trials [77,78]; note that it did not produce further improvement in weight in patients already on diet and exercise programs [79]. Irrespective of weight loss, metformin does ameliorate glucose tolerance and reduce insulin levels when associated with lifestyle modification [80]; however, these positive effects are rapidly lost within the first year after its discontinuation, stressing again the need to adopt healthy habits in order to reduce long-term metabolic complications [81,82]. 

Metformin is potentially able to improve lipid profile as well by increasing HDL-C and reducing TG levels, both known predictors of CVD, through its direct and indirect effects on the hepatic metabolism of free fatty acids and on hyperinsulinemia respectively [83]. However, TC levels remained unchanged in most cases [84]. Moreover, metformin use appears to improve inflammatory markers that are associated with increase CV risk, such as PAI-1, endothelin, CRP, advanced glycation end products (AGEs) soluble markers of inflammation and endothelial dysfunction irrespective of insulin sensitization [7]. 

The benefit of metformin treatment goes beyond the effects on cardio-metabolic impairment thanks to its pleiotropic actions on several insulin-sensitive tissues, such as the liver, skeletal muscles, adipose tissue, endothelium, and ovaries; thus, also targeting the reproductive abnormalities that characterize the syndrome [85]. Although not as effective as oral contraception in restoring menstrual regularity, metformin can represent a valuable alternative in those women in whom COC treatment is contraindicated [2]. Besides, metformin was found to significantly raise the number of spontaneous and citrate clomiphene-induced ovulations, pregnancy rate, and improve outcomes [76]. 

The starting dose is usually 500 mg/day to be progressively titrated to 2000 mg/day to reduce gastrointestinal symptoms, the most common cause of treatment withdrawal. Extended-release formulas can improve gastrointestinal tolerability [86], thus enhancing long-term compliance. 

### 5.2. Inositols

Inositols are cyclic polyalcohols that occur naturally in nine stereoisomeric forms. Among them, myo-inositol (MYO) is the most abundant in humans, primarily derived from dietary sources [87]. MYO can be converted to D-chiro-inositol (DCI), the second most abundant form, by an epimerase enzyme stimulated by insulin. As a result, every organ and tissue can balance inositol levels and DCI/MYO ratio, regulating several physiological processes, such as ion channel permeability, cytoskeleton remodeling, developmental processes, stress response, and endocrine modulation. In particular, both MYO and DCI act as inositolphosphoglycans (IPGs), which act as intracellular second messengers in insulin signaling. Specifically, MYO regulates cellular uptake and use of glucose, and its concentration is elevated in the brain and in the heart; on the contrary, DCI controls glycogen synthesis and it is more abundant in tissues like liver, fat, and muscles [88]. In the ovary, MYO is involved in glucose uptake, but it also acts as second messenger of FSH signaling, while DCI is involved in the insulin-mediated androgen synthesis [89]. 

Previous studies demonstrated alterations in tissue availability or metabolism of IPGs in PCOS that could be involved in PCOS onset and development [90]. PCOS patients have increased urinary clearance of DCI and decreased glucose-stimulated release of DCI-IPG mediator [91,92], which could contribute to IR and compensatory hyperinsulinemia [93]. In 1999, Nestler et al. were the first to report DCI efficacy in the treatment of obese PCOS women for 6–8 weeks, demonstrating improved insulin sensitivity, ovulatory function, and hyperandrogenism and decreased blood pressure levels and plasma TG concentrations [94]. All these effects have been subsequently confirmed in lean PCOS women [95]. MYO supplementation also improved metabolic and oxidative imbalances and it was even effective at restoring regular menses and ovulation [96,97,98].

Additional studies showed that MYO concentration in the follicular fluid was directly correlated with high oocyte quality [99] and that MYO supplementation improved oocyte and embryo quality and reduced the amount of recombinant FSH administered during ovarian stimulation protocols [100,101]. By contrast, high doses of DCI alone were found significantly detrimental for oocytes and fertility in PCOS women [102]. Unlike the liver and the muscles, ovaries maintain normal insulin sensitivity [103]; therefore, it has been proposed that the hyperinsulinemia secondary to IR in PCOS women enhances ovarian MYO to DCI epimerization, causing an increase DCI/MYO ratio, called the “the DCI ovarian paradox” [104]. The consequent MYO deficiency in the ovary might be involved in the impaired FSH signaling and poor oocyte quality observed in PCOS [89]. Subsequent studies suggested that MYO plus DCI administered in the 40:1 ratio restores ovarian function and improves the metabolic profile in PCOS women [105], although this concept is controversial [106]. Furthermore, possible differences, if any, between DCI and MYO as for cardiometabolic consequences remain unclear. Some studies reported a reduction of blood TG and TC as well as an increase of HDL-C [107], while effects on weight are conflicting [90].

A recent meta-analysis of six clinical trials proved that short-term treatment with metformin or MYO has comparable hormonal and metabolic effects on PCOS women and treatment with MYO is better tolerated than that with metformin [108]. Considering that the effects of metformin on IR is in part due to increasing the DCI-IPG release from cell membranes, mediating intracellular insulin signaling [93], metformin treatment combined with MYO and/or DCI supplementation could have a synergistic effect and allow reducing metformin dosages, useful for intolerant patients. A recent randomized clinical trial seems to confirm that this combination leads to greater improvement in metabolic profile and ovarian function compared with metformin alone [109]. The beneficial effects of MYO were also demonstrated in teenagers affected by PCOS, and the combination of MYO with COCs showed a better metabolic profile and a greater antiandrogenic effect than the use of COCs alone [110].

Other studies evaluated the association on DCI or MYO with different compounds that could enhance their therapeutic effect. For example, the association of inositols with alpha-lipoic acid, a biological antioxidant and natural cofactor of mitochondrial dehydrogenase complexes, demonstrated a significant improvement of body mass index (BMI), glucose, and lipid metabolism and endocrine and reproductive abnormalities in heterogeneous groups of PCOS women [111,112,113]. Even the association with folic acid (vitamin B9) was found to be effective in improving metabolic profile, especially preventing hyperhomocysteinemia, a CVD risk factor, frequently occurring in PCOS women or induced by metformin treatment [105,114].

In conclusion, based on available data, currently inositol therapy is widely accepted as a safe, effective, and integrative treatment for PCOS patients, improving several hormonal and metabolic alterations typical of the syndrome. Due to its tolerability, it could be associated with metformin and/or COCs, to reduce adverse effects and have a synergistic action. However, further studies on larger cohorts and with greater statistical power are needed to clarify outcomes and suitable therapeutic regimens for DCI and MYO.

### 5.3. Acarbose

Acarbose is an alpha-glucosidase inhibitor that reduces glucose intestinal absorption of glucose, with a consequent reduction of the postprandial wave and insulin peak. The ACE study conducted in patients with impaired glucose tolerance and coronary heart disease demonstrate that acarbose did not reduce the incidence of CV events but delayed the progression to T2DM [115]. This effect might indirectly reduce the risk of CVD disease in the long run, which is of particular interest in PCOS patients. Moreover, acarbose does not induce hypoglycemic events, making it suitable for PCOS patients [116]. However, data on PCOS are scant due to the frequent gastrointestinal side effects associated with acarbose use that limit its long-term assumption. In the available studies, acarbose dose ranged from 150 to 300 mg/day and was given in monotherapy or in combination with metformin. The limited evidence suggests that acarbose is able to reduce total testosterone levels and improve lipid profile by decreasing TG and VLDL-C, and increasing HDL-C levels in PCOS patients [117,118]. Some studies also reported a significant decrease in BMI especially in obese patients, whereas data on IR were not assessed in most of them [119,120]. The small number of patients included in these studies are not sufficient to draw conclusions on any cardiovascular benefits of acarbose [116]. 

### 5.4. Thiazolidinediones

Thiazolidinediones (TZDs) act as peroxisome proliferator-activated receptor-gamma (PPAR-γ) agonists. These agents increase insulin sensitivity by acting on muscle and liver to increase glucose utilization and decrease glucose production. Since troglitazone was withdrawn from the market, as hepatotoxicity and rosiglitazone use was restrained due to excess CV events, most data in PCOS women regards pioglitazone [75]. 

Taken as add-on therapy at 45 mg/day to diet and metformin when not effective, pioglitazone was able to significantly improve insulin, glucose, and IR secretion; when DHEAS fell, HDL-C and SHBG rose, and menstrual regularity was achieved in 13 PCOS patients with no side effects [121,122]. In a randomized placebo-controlled trial on 40 premenopausal women, pioglitazone was more effective than placebo in improving insulin sensitivity, hyperandrogenism, and ovulation rate [123]. A head-to-head comparison on 57 PCOS patients treated with either metformin (850 mg three times a day) or pioglitazone (30 mg/day) showed no differences in term of improvement in IR and hirsutism, even though pioglitazone was associated with increase in BMI and waist-to-hip ratio [124]. These paradoxical results can be explained by the beneficial shift from abdominal to subcutaneous fat and the simultaneous improvement in insulin sensitivity induced by TZD, but further studies are required to confirm this hypothesis. As a collateral outcome, eight pregnancies occurred within the 6 months of treatment, five in pioglitazone arm and three in metformin one, suggesting that pioglitazone ameliorates menstrual cycles and ovulation better than metformin, whereas it is less effective in reducing body weight and hirsutism [123,124,125,126]. 

Last, pioglitazone can also act on androgen excess through its unique mechanism of action. Besides its androgen-lowering effects due to inhibition of P450c17 and 3β-hydroxysteroid dehydrogenase [127], two key enzymes in human androgen synthesis, pioglitazone increases level of serum SHBG with consequent decrease in the free androgen. Given the safety concerns regarding teratogenicity, fluid retention and heart failure; however, TZD use is not recommended in PCOS patients [75].

### 5.5. Incretin Mimetics

Glucagon-like peptide-1 (GLP-1) is an incretin hormone produced in the digestive tract in response to food intake that stimulates insulin and inhibits glucagon secretion in a glucose-dependent manner. The half-life of GLP-1 is primarily regulated by dipeptidyl peptidase (DPP) 4, a serine protease expressed in several tissues, including the liver, intestine, and endothelial cells that regulates both GLP-1 and glucose-dependent insulinotropic polypeptide (GIP) degradation [128]. GLP-1 receptor agonists (RAs) were developed for treatment of hyperglycemia in T2DM patients; the LEADER study reported that liraglutide improved CVD outcomes in high-risk diabetic patients with previous CV events through different pathways [129]. Beyond glycemic reduction and CV prevention, GLP-1RAs produce a significant decrease in body weight and may be therefore of interest in the management of obese PCOS women, even though they are not yet approved for this indication. 

To date, GLP-1RAs have been used in small clinical trials in PCOS patients with encouraging results. Overall, the weight loss in PCOS was consistent with previous findings on diabetic patients [130,131]. Liraglutide 1.2 mg/day associated with metformin 1000 mg bid was found to be effective in reducing BMI and waist circumference particularly in obese PCOS patients poor responsive to metformin monotherapy, in particular between 8 and 12 weeks of treatment [130,132,133]. As a result of weight loss, liraglutide seems to ameliorate the coagulative profile, especially promoting fibrinolysis; thus, suggesting a possible beneficial effect on both venous thrombosis and CV markers [134,135]. Still, this 26-week-long study was way too short to confirm the protective role of GLP-1RAs in PCOS patients, and larger studies are needed [135]. As they are rather young, most PCOS patients usually do not present with overt CVD; therefore, early biomarkers of endothelial dysfunction have been assessed instead of major CV outcomes such as myocardial infarction or stroke. The highest liraglutide dosage (3 mg/day) available for obesity treatment was also assessed in PCOS patients, resulting in greater effect on weight loss and visceral adiposity than lower liraglutide doses combined with metformin, but with similar effect on glucose homeostasis [136]. The combination of exenatide 10 µg bid and metformin 1000 mg bid was demonstrated to be superior to each single agent alone in improving not only BMI, central adiposity, and IR indexes, but also menstrual cycles and hirsutism over a treatment period of 24 weeks, against only a slight increase in gastrointestinal side effects [137]. Other studies confirmed the beneficial effect of exenatide on metabolic and anthropometric parameters [138,139,140], whereas data on inflammatory markers are still controversial [134,137,138]. Weekly GLP1-RAs (dulaglutide and semaglutide) are also available for T2DM treatment, but they have not been used in PCOS so far. Dulaglutide could be of particular interest as it proved to be effective in primary CV prevention in diabetic patients with CV risk factors [141]. Due to the short course of GLP-1RAs, their durability in this specific clinical setting needs to be further investigated. 

To date, fewer data are available on DPP4-inhibitors (DPP4-Is) in PCOS. These oral agents reduce GIP and GLP-1 degradation, thus increasing their bioavailability. The effect of saxaglitptin alone (5 mg/day) or in combination with metformin (2000 mg/day) was compared to metformin treatment alone in a randomized open-label study on diabetic PCOS patients; each group of treatment reached a significant reduction of surrogate markers of CV and metabolic risk such as high-sensitivity CRP, weight, and HOMA index with no differences between groups, with the exception of a significant glycated hemoglobin A1c (HbA1c) reduction in the combination group [142]. Sitagliptin was also tested in association with metformin to prevent weight gain in obese PCOS patients previously treated with liraglutide; the combination of the two drugs was more efficient than metformin monotherapy [143]. Despite DPP4-Is’s neutral effect on weight, as also demonstrated in T2DM trials [144], a short course of 100 mg sitagliptin was effective in reducing visceral adiposity in PCOS women [145]. DPP4-Is can be therefore considered as an alternative option in metformin-intolerant women with IGT or T2DM due to their safety profile and convenient administration route [146]. As observed for other treatment options, available studies on DPP4-i in PCOS are too short to determine the long-term effects of this treatment in term of prevention from CV events. A future perspective in PCOS involves the combined GLP-1/GIP receptor agonist that is currently under investigation for T2DM patients.

### 5.6. SGLT2 Inhibitors

Sodium–glucose co-transporter type 2 (SGLT2) inhibitors are the newest class of oral antidiabetic drug developed; they induce glycosuria by inhibiting glucose reuptake in the renal proximal tubule and thus lower blood glucose. As a result of the increase in urinary glucose excretion, SGLT2-Is induce significant caloric loss and body weight reduction [147]. Furthermore, reduction of arterial pressure due to osmotic diuresis was reported. The EMPA-REG and CANVAS trials proved for empagliflozin and canagliflozin, respectively, their CV safety in T2DM patients as a decrease of both CV deaths and the incidence of myocardial infarction and stroke was observed [148,149]. 

Empagliflozin was tested over a 12-week period of treatment in a randomized open-label trial in overweight and obese PCOS patients. This medication was more effective than standard slow-release metformin treatment (500 mg three times a day) in reducing weight, BMI, waist and hip circumferences, and total body fat. No changes in androgen levels were observed [150]. Empagliflozin was well tolerated and no drug-related adverse events were recorded. Effects on ovulation and menstrual frequency have not been assessed yet [150]. Given the known protective CV action of SGLT2-Is, markers of endothelial dysfunction were assessed by the same authors. Surprisingly, both empagliflozin and metformin led to an increase in markers of endothelial activation; however, it remains unclear whether these changes represent a transitory, adaptive response to regenerate the endothelium, limit vascular damage, and restore homeostasis, or if they contribute to endothelial dysfunction and increased CVD risk in the long term [151]. 

More randomized controlled trials are needed to confirm CV protection of SGLT2-Is in PCOS patients.

## 6. Metabolic Impact of Combined Oral Contraceptives 

Combined oral contraceptives (COCs) are recommended by Endocrine Society guideline as first-line treatment for menstrual abnormalities and hyperandrogenic disorders in PCOS women not seeking fertility [75]. However, their use has been associated with the onset of glucose intolerance, hypertension, hypertriglyceridemia, and elevated CRP levels in healthy women [152,153]. For these reasons, the use of COCs has been suggested to be an additional cardiovascular risk factor in PCOS, given the high prevalence of IR, central obesity, diabetes, hypertension, and dyslipidemia in these patients [154]. By contrast, COC treatment may improve cardiometabolic risk, lowering hyperandrogenism and its negative effect on IR and fat distribution [155].

At present, the effects of COCs on cardiometabolic aspects are still controversial, because available studies are mainly small, short-term, and very heterogeneous due to several COC formulations used [152,153]. Indeed, the combination of different types and doses of estrogen with the type of progestin used may strongly influence not only hormonal findings but also thromboembolic and cardiometabolic risk, which also varies according to age, body type, and genetic predisposition of the patient. 

It is well recognized that the risk of venous thromboembolism with the use of COCs in the general population increases with increasing doses of 17α-ethynylestradiol (EE) and the use of progestins with antiandrogenic activity [156]. EE induces several liver proteins, such as sex hormone binding globulins (SHBG) and coagulations factors, and this effect can be counteracted by the androgenic properties of progestins associated with EE [157]. This balance also affects the metabolic profile. Estrogen, especially EE, impairs insulin action in a dose-dependent manner [158], in particular in obese patients [159]. However, the same dose of EE can have a different metabolic impact according to the associated progestin. 

Cyproterone acetate (CPA) acts as a partial agonist at androgen receptors (AR), thus it inhibits DHT binding to the intracellular AR and therefore its nuclear translocation. Furthermore, CPA appears to induce a direct inhibiting effect on 5-alpha-reductase. A meta-analysis including non-obese PCOS women treated for 3–12 months with COCs, mainly containing CPA, reported non-significant change in fasting glucose and insulin levels and IR after treatment [160]. A subsequent randomized controlled trial in overweight/obese PCOS women evidenced worsening glycemic and lipid profile after 16 weeks of treatment with only a COC containing low-dose EE and norethindrone acetate, an androgenic progestin; however, these effects could be prevented by concomitant lifestyle intervention [161]. Another trial demonstrated that lean PCOS women showed a reduction of total and abdominal fat when switched to a COC based on drospirenone (DSP) instead of gestodene [162]. DSP is a synthetic analog of SPA with a similar pharmacological profile to endogenous progesterone. It has anti-mineralocorticoid properties [163]: the affinity of DSP for MR is about five times as high as that of aldosterone itself. In addition, DSP has some antiandrogen action: its potency is about 30% of that of CPA, the most potent antiandrogenic progestin. A recent randomized trial confirmed that COCs including DSP have more favorable effects on IR, lipid profiles, and high sensitivity C-reactive protein compared with COCs with clormadinone in PCOS women [164]. Collectively, desogestrel- and norgestimate-based COCs induce a small but nonsignificant increase in fasting insulin levels in PCOS patients [153].

Considering the lipid profile, estrogen increases both serum TG and HDL-C levels, without a significant impact on LDL-C levels [153,165]. This effect seems to be less sensitive to the residual androgenic properties of progestins. Therefore, every COC with EE, regardless of progestin preparation, tends to increase total cholesterol and TG levels in PCOS patients. Considering that PCOS women frequently present increased TG and low HDL-C levels, associated with IR, obesity and hyperandrogenism [5], the Endocrine Society guidelines recommend screening all PCOS women routinely for dyslipidemia [75]. 

Available studies are still controversial about the impact of different COCs, even with antiandrogen progestins, on HbA1c levels, BMI, low-grade inflammation, and adipose tissue distribution [152,153]. However, a recent systematic review and metanalysis of available randomized control trials suggests that addition of metformin to COCs provided further benefits on fasting glucose levels, insulin sensitivity, and BMI with respect to COC therapy alone in adult PCOS patients [165]. Preliminary data seem to confirm that even addition of MYO to COCs could provide a better metabolic profile than COCs alone in adolescents with PCOS [110].

Another important aspect that could be adversely influenced by COCs is hypertension. EE increases angiotensinogen synthesis by activating the renin–angiotensin–aldosterone system (RAAS) and inducing the onset of sodium and water retention, hypertension, and cardiometabolic alterations [166]. Actually, COC-induced hypertension has become a rare side effect after the introduction of COCs with low EE doses (20–30 µg) [167]. However, PCOS women show higher aldosterone levels compared with age- and BMI-matched healthy controls that correlate with blood pressure values as well as metabolic and cardiovascular markers [168,169]. Moreover, hypertension has been associated with IR, obesity, hyperandrogenism, and aging [170]. Therefore, COC use should be carefully evaluated at baseline condition after screening PCOS women for hypertension and pre-existing metabolic and cardiovascular risk factors [75]. Even during follow-up, this treatment should be confirmed only after exclusion of COC-related side effects or risk factors. The addition of spironolactone to COCs has been proposed not only to enhance the antiandrogen effects, but also to reduce short- and long-term side effects related to RAAS activation [58]. 

Finally, two recently developed COCs with natural estrogen (17β-estradiol and 17β-estradiol valerate) could be a valid therapeutic alternative in PCOS. Indeed, these estrogenic agents undergo less hepatic metabolism than EE and have a more favorable effect on metabolic profile and hemostatic balance, without blood pressure changes [171,172]. Preliminary data confirm that the COC with 17β-estradiol valerate and dienogest could exert a positive effect in reducing hyperandrogenism and IR in PCOS women, without BMI and glucose metabolism changes [173,174]. Dienogest is a third-generation C19-norprogestin with antiandrogenic properties, which is more potent and has less androgenic action than second-generation agents. However, further studies are needed to confirm the possible role of these new COCs with natural estrogen in preventing the main endocrine and cardiometabolic features of PCOS.

In conclusion, COCs are the first-line treatment for PCOS women not seeking fertility, and all formulations appear to be equally effective for hyperandrogenic disorders [75,175]. Progestins such as drospirenone and dienogest appear to mediate the most significant anti-androgenic clinical effects on symptomatic PCOS women owing to their pharmacological characteristics. However, given the heterogeneous and chronic nature of PCOS, the type of estroprogestin could affect thromboembolic and cardiometabolic risk. Therefore, both at baseline and during follow-up it is fundamental to consider not only patients’ requests, but also the cardiometabolic and thromboembolic risk factors that could contraindicate treatment with COCs. The use of COCs containing androgenic progestins, such as levonorgestrel, should be considered to minimize thromboembolic risk in PCOS patients, for example, who are obese or over 35 years of age. In this context, initial treatment with COCs with low-dose EE (usually 20 µg) or natural estrogen could help control both menstrual and hormonal disorders without worsening the cardiometabolic profile. Finally, diet and lifestyle changes should be always provided to PCOS patients and combined with pharmacological treatments, such as insulin sensitizers, which could be associated with COCs to yield a synergistic and comprehensive effect on the multiple features of the syndrome.

## 7. Statins

Small intervention studies have shown some benefit of statin treatment in endocrinological and metabolic endpoints in PCOS women [176,177,178]. These agents have been proposed as a therapeutic option for reducing testosterone levels, either alone or in combination. Compared with other therapeutic strategies, atorvastatin showed greater reduction in testosterone levels in a recent metanalysis [179]. In addition, statins improve lipid profiles and ameliorate inflammatory reactions [180]. Notably, atorvastatin treatment has been shown to improve adipocyte function in PCOS women. PCOS is associated with low-grade inflammation of adipose tissue as well as complement activation. In particular, markers of adipose tissue dysfunction and inflammation, namely, acylation-stimulating-protein, interleukin-6, and monocyte-chemoattractant-protein-1, were significantly decreased following 12 weeks of atorvastatin 20 mg/day in 20 medication-naïve obese women with PCOS with respect to placebo [181]. Despite these encouraging findings, the role of statins in PCOS management requires further validation.

Table 1 below summarizes the options for pharmacological PCOS management as discussed in the above sections.

## 8. Conclusions

Hyperandrogenism is a hallmark of PCOS. Androgens in women play an unfavorable role on the CV system and, together with hyperinsulinemia and/or obesity, can cause high blood pressure, increase of inflammatory markers, dyslipidemia, and CV events in PCOS patients. As these pathologic alterations or diseases usually begin in later periods of life, such as the third or fourth decade, when there may already be irreversible damage, overweight, obesity, hyperinsulinemia, IR, and dyslipidemia should represent the first step of therapeutic approach in PCOS subjects. However, these conditions represent co-morbidities which are not always present, especially in adolescence, while hyperandrogenism is the most frequent and one of the earliest causes of disease in PCOS. Thus, early correction of hyperandrogenism, already in youth, could improve or even prevent the aforementioned co-morbidities in later life. COCs represent for many scientific societies the drug of choice for the correction of hyperandrogenism in PCOS. However, COC intake can be accompanied by thromboembolic events, weight gain, and possibly additional, yet less frequent adverse effects. Thus, COC treatment needs to be personalized considering the metabolic and cardiovascular risk profile and family history of individual patients [182,183]. Furthermore, metabolic abnormalities impact on global cardiovascular risk and mortality rate in the adult age of patients affected by PCOS [184,185]. Therefore, medications that can be taken for long periods with an acceptable safety profile should be considered. Antiandrogens, oral hypoglycemic agents, and possibly statins represent additional therapeutic options for treatment of symptoms of androgen excess in PCOS and control of cardiometabolic risk factors. A sometimes-difficult balance between efficacy and safety of available medications according to disease phenotypes has to be found in individual women.

## Figures and Tables

**Figure 1 ijms-21-09554-f001:**
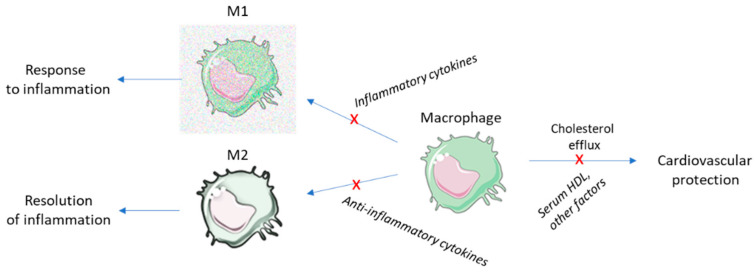
Alterations in macrophage functional phenotypes in PCOS. Macrophages are plastic cells that respond to pro- and anti-inflammatory stimuli to adjust to environmental changes and maintain homeostasis. Compared with healthy women, blood-derived macrophages from PCOS women are impaired in their response to cytokines, suggesting a reduced capacity to adjust to microenvironmental changes in different conditions. In addition, the serum from PCOS women is less effective in promoting cholesterol efflux from macrophage cell models with respect to that from healthy women. Cholesterol efflux is driven by serum HDL subfractions but also by other cellular and circulating factors. Impaired cholesterol efflux capacity is associated with an increased risk for atherothrombotic events. X: impairment observed in PCOS women vs. controls.

**Table 1 ijms-21-09554-t001:** A summary of pharmacological management of cardiometabolic risk factors in PCOS women.

Drug Class	Mechanism	Endpoints	Side Effects
Antiandrogens	Androgen receptor antagonists5 alpha reductase inhibitors	Improved lipid profile and insulin resistanceVascular compliance	Dyslipidemia (BCL) possibly due to concomitant medication
Oral hypoglycemics	Insulin sensitization	Weight loss Improvement in insulin sensitivity resulting in lower circulating androgen concentration	Weight gain (TZD)
Combined oral contraceptives	Steroid hormone receptor modulation	Lower androgen levels	DyslipidemiaThrombogenic risk (less with antiandrogenic progestins)
Statins	Cholesterol biosynthesis inhibition	Reduced testosterone levels and inflammatory markers

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
