# Peer review of "Pharmacological Approaches to Controlling Cardiometabolic Risk in Women with PCOS"

_ijms, 2020, doi:10.3390/ijms21249554_

Round 1

Reviewer 1 Report

The paper assesses the correlation between different therapies in PCOS subjects in relation to cardiometabolic risk in an extremely in-depth manner.

Lifestyle changes must be the basis of any treatment because, if effective, it could significantly reduce cardiometabolic risks in later life

Furthermore, as also specified by the authors, each treatment must be personalized.

Author Response

We thank the Reviewer for these comments and suggestions. The importance of lifestyle changes has been highlighted on p. 2, lines 54-55 of the revised manuscript.

Reviewer 2 Report

Dear Editors and authors,

I am grateful to revise this article.

It reports an interesting review about the “Pharmacological approaches to controlling cardiometabolic risk in
women with PCOS”

In my opinion, the authors should revise their paper, considering the following comments to ameliorate the text:

  • Abstract should contain a brief introduction to the study and should be divided into Introduction, material and methods, results and conclusios.
  • The manuscript is incomplete since the section regarding the punctual description of materials and methods is lacking.
  • Introduction:
  • Line 54-56: I suggest to add a reference; in particular, it could be useful to cite the guidelines by Teede et al. (Teede HJ, Misso ML, Costello MF, Dokras A, Laven J, Moran L, Piltonen T, Norman RJ; International PCOS Network. Recommendations from the international evidence-based guideline for the assessment and management of polycystic ovary syndrome. Fertil Steril. 2018 Aug;110(3):364-379. doi: 10.1016/j.fertnstert.2018.05.004. Epub 2018 Jul 19. PMID: 30033227; PMCID: PMC6939856.)
  • Line 68: authors should insert an appropriate reference as for involvement of genetics in potential cardiovascular risk of PCOS women.
  • Line 128: authors should add a reference about “the role of hyperandrogenism in dyslipidemia, diabetes and CVD”.
  • Line 135: in real clinical practise, this concept appears less clear considering the recognized psychological consequences of dermatological signs associated with higher levels of androgens. The authors should clarify this aspect. (See also De Niet JE et al Psychological well-being and sexarche in women with polycystic ovary syndrome Hum Reprod; Hahn S, Janssen OE, Tan S, Pleger K, Mann K, Schedlowski M, Kimmig R, Benson S, Balamitsa E, Elsenbruch S. Clinical and psychological correlates of quality-of-life in polycystic ovary syndrome. Eur J Endocrinol. 2005 Dec;153(6):853-60. doi: 10.1530/eje.1.02024. PMID: 16322391)
  • Line 258, 414, etc: in my opinion, the chapter regarding ketoconazole, thiazolidinediones, incretin mimetics and SGLT2 inhibitors, could be shortened since their use for patients affected by PCOS is limited.
  • Line 365: in my opinion, the authors should not emphasize the concept of “40:1 ratio” because it is still controversial. (See also Scambia G. et al The role of Inositols in PCOS – Opinion Paper Italian Journal of Gynaecology and Obstetrics2019 )
  • Line 386: the authors could insert other significant evidence regarding the synergistic role of alpha-lipoic acid (See also Fruzzetti F, Capozzi A, Canu A, Lello S. Treatment with d-chiro-inositol and alpha lipoic acid in the management of polycystic ovary syndrome. Gynecol Endocrinol. 2019 Jun;35(6):506-510. doi: 10.1080/09513590.2018.1540573. Epub 2019 Jan 7. PMID: 30612488 ; Genazzani AD, Prati A, Marchini F, Petrillo T, Napolitano A, Simoncini T. Differential insulin response to oral glucose tolerance test (OGTT) in overweight/obese polycystic ovary syndrome patients undergoing to myo-inositol (MYO), alpha lipoic acid (ALA), or combination of both. Gynecol Endocrinol. 2019 Dec;35(12):1088-1093. doi: 10.1080/09513590.2019.1640200. Epub 2019 Jul 13. PMID: 31304823.)
  • Line 607: however, the authors should underline that progestins, such as drospirenone and dienogest, appear to mediate the most favorable antiandrogenic effects owing to their pharmacological properties.
  • Discussion
  • Line 651: the authors should notice that treatment with EPs need to be customize taking into account cardiovascular risk profile and family history of each patients. (See also Fruzzetti F, Cagnacci A. Venous thrombosis and hormonal contraception: what's new with estradiol-based hormonal contraceptives? Open Access J Contracept. 2018 Nov 8;9:75-79. doi: 10.2147/OAJC.S179673. PMID: 30519125; PMCID: PMC6239102.; De Leo V, Musacchio MC, Cappelli V, Piomboni P, Morgante G. Hormonal contraceptives: pharmacology tailored to women's health. Hum Reprod Update. 2016 Sep;22(5):634-46. doi: 10.1093/humupd/dmw016. Epub 2016 Jun 15. PMID: 27307386.)
  • Another important aspect that authors should report in the conclusive section is the effective impact of metabolic abnormalities on global cardiovascular risk and mortality rate in the adult age of patients affected by PCOS (See Rizzo M, Berneis K, Spinas G, Rini GB, Carmina E. Long-term consequences of polycystic ovary syndrome on cardiovascular risk. Fertil Steril. 2009 Apr;91(4 Suppl):1563-7. doi: 10.1016/j.fertnstert.2008.09.070. Epub 2008 Oct 30. PMID: 18976752; Pierpoint T, McKeigue PM, Isaacs AJ, et al. Mortality of women with polycystic ovary syndrome at long-term follow-up. J Clin Epidemiol 1998;51:581–6).
  • English form need to be revised.

Best regards

Author Response

We thank the Reviewer for providing helpful comments and suggestions.

  • Abstract should contain a brief introduction to the study and should be divided into Introduction, material and methods, results and conclusios.
  • The manuscript is incomplete since the section regarding the punctual description of materials and methods is lacking.

Because this is review article and no experimental results are provided, we thought to write a single-paragraph abstract and skip materials and methods.

  • Introduction:
  • Line 54-56: I suggest to add a reference; in particular, it could be useful to cite the guidelines by Teede et al. (Teede HJ, Misso ML, Costello MF, Dokras A, Laven J, Moran L, Piltonen T, Norman RJ; International PCOS Network. Recommendations from the international evidence-based guideline for the assessment and management of polycystic ovary syndrome. Fertil Steril. 2018 Aug;110(3):364-379. doi: 10.1016/j.fertnstert.2018.05.004. Epub 2018 Jul 19. PMID: 30033227; PMCID: PMC6939856.)

This reference has been cited (line 55 of the revised manuscript).

  • Line 68: authors should insert an appropriate reference as for involvement of genetics in potential cardiovascular risk of PCOS women.

A reference has been inserted (line 67 of the revised manuscript).

  • Line 128: authors should add a reference about “the role of hyperandrogenism in dyslipidemia, diabetes and CVD”.

A reference has been added (lines 126-127 of the revised manuscript).

  • Line 135: in real clinical practise, this concept appears less clear considering the recognized psychological consequences of dermatological signs associated with higher levels of androgens. The authors should clarify this aspect. (See also De Niet JE et al Psychological well-being and sexarche in women with polycystic ovary syndrome Hum Reprod; Hahn S, Janssen OE, Tan S, Pleger K, Mann K, Schedlowski M, Kimmig R, Benson S, Balamitsa E, Elsenbruch S. Clinical and psychological correlates of quality-of-life in polycystic ovary syndrome. Eur J Endocrinol. 2005 Dec;153(6):853-60. doi: 10.1530/eje.1.02024. PMID: 16322391)

This aspect has been clarified (lines 134-136 of the revised manuscript).

  • Line 258, 414, etc: in my opinion, the chapter regarding ketoconazole, thiazolidinediones, incretin mimetics and SGLT2 inhibitors, could be shortened since their use for patients affected by PCOS is limited.

The Sections mentioned by the Reviewer have been shortened as recommended.

  • Line 386: the authors could insert other significant evidence regarding the synergistic role of alpha-lipoic acid (See also Fruzzetti F, Capozzi A, Canu A, Lello S. Treatment with d-chiro-inositol and alpha lipoic acid in the management of polycystic ovary syndrome. Gynecol Endocrinol. 2019 Jun;35(6):506-510. doi: 10.1080/09513590.2018.1540573. Epub 2019 Jan 7. PMID: 30612488 ; Genazzani AD, Prati A, Marchini F, Petrillo T, Napolitano A, Simoncini T. Differential insulin response to oral glucose tolerance test (OGTT) in overweight/obese polycystic ovary syndrome patients undergoing to myo-inositol (MYO), alpha lipoic acid (ALA), or combination of both. Gynecol Endocrinol. 2019 Dec;35(12):1088-1093. doi: 10.1080/09513590.2019.1640200. Epub 2019 Jul 13. PMID: 31304823.)

References have been added as suggested (lines 376-379 of the revised manuscript).

  • Line 607: however, the authors should underline that progestins, such as drospirenone and dienogest, appear to mediate the most favorable antiandrogenic effects owing to their pharmacological properties.

We thank the Reviewer for this comment that has been incorporated into the revised manuscript (lines 585-587).

  • Discussion
  • Line 651: the authors should notice that treatment with EPs need to be customize taking into account cardiovascular risk profile and family history of each patients. (See also Fruzzetti F, Cagnacci A. Venous thrombosis and hormonal contraception: what's new with estradiol-based hormonal contraceptives? Open Access J Contracept. 2018 Nov 8;9:75-79. doi: 10.2147/OAJC.S179673. PMID: 30519125; PMCID: PMC6239102.; De Leo V, Musacchio MC, Cappelli V, Piomboni P, Morgante G. Hormonal contraceptives: pharmacology tailored to women's health. Hum Reprod Update. 2016 Sep;22(5):634-46. doi: 10.1093/humupd/dmw016. Epub 2016 Jun 15. PMID: 27307386.)

We thank the Reviewer for this comment that has been incorporated into the revised manuscript (lines 630-631).

  • Another important aspect that authors should report in the conclusive section is the effective impact of metabolic abnormalities on global cardiovascular risk and mortality rate in the adult age of patients affected by PCOS (See Rizzo M, Berneis K, Spinas G, Rini GB, Carmina E. Long-term consequences of polycystic ovary syndrome on cardiovascular risk. Fertil Steril. 2009 Apr;91(4 Suppl):1563-7. doi: 10.1016/j.fertnstert.2008.09.070. Epub 2008 Oct 30. PMID: 18976752; Pierpoint T, McKeigue PM, Isaacs AJ, et al. Mortality of women with polycystic ovary syndrome at long-term follow-up. J Clin Epidemiol 1998;51:581–6).

We thank the Reviewer for this comment that has been incorporated into the revised manuscript (lines 631-633).

  • English form need to be revised.

Several edits have been made throughout the text.

Round 2

Reviewer 2 Report

Dear Editors and authors,

I appreciated the revised version of this paper.

Nevertheless, in my opinion, the authors could furtherly enhance the scientific impact of their review including the following suggestions:

  • Materials and methods: even if in a short form, a section regarding data collection and methodology should be always present in any article, also for a review article.
  • Main text:
  • Line 198, etc: in my opinion, the chapters regarding bicalutamide, ketoconazole and incretin-mimetics could be furtherly shortened since their use in patients affected by PCOS is still limited and not approved. The Authors should take into account the latter concept and they should underline this aspect in their narrative review for each therapy which use is still off-label.
  • Line 367-369: in my opinion, the authors should not emphasize the concept of “40:1 ratio” because it is a matter of debate. In fact, “the DCI ovarian paradox” is not largely approved from a pharmacological and clinical point of view as suggested by the previously attached reference (Scambia G. et al The role of Inositols in PCOS – Opinion Paper Italian Journal of Gynaecology and Obstetrics2019). Hence, the authors should better argument their statements and/or highlight the possible differences, if any, between DCI and MYO as for cardio-metabolic consequences.
  • Line 619: the authors should underline that progestins, such as drospirenone and dienogest, appear to mediate the most significant “anti-androgenic” rather “metabolic” clinical effects on symptomatic PCOS women owing to their pharmacological characteristics.
  • Line 632: in my opinion spironolactone should be deleted, since its synergistic role in PCOS treatment maybe argued.

Sincerely

Author Response

We thank the Reviewer for further comments and suggestions.

  • Materials and methods: even if in a short form, a section regarding data collection and methodology should be always present in any article, also for a review article.

A short Materials and methods section has been added (revised pdf manuscript, page 2, lines 56-60).

  • Line 198, etc: in my opinion, the chapters regarding bicalutamide, ketoconazole and incretin-mimetics could be furtherly shortened since their use in patients affected by PCOS is still limited and not approved. The Authors should take into account the latter concept and they should underline this aspect in their narrative review for each therapy which use is still off-label.

We have further shortened the sections on bicalutamide and ketoconazole as recommended and mentioned off-label use where indicated. With regard to incretin mimetics, we made less changes as all reviewed studies actually focus on PCOS women and CV risk factors. We highlighted the off-label use though.

  • Line 367-369: in my opinion, the authors should not emphasize the concept of “40:1 ratio” because it is a matter of debate. In fact, “the DCI ovarian paradox” is not largely approved from a pharmacological and clinical point of view as suggested by the previously attached reference (Scambia G. et al The role of Inositols in PCOS – Opinion Paper Italian Journal of Gynaecology and Obstetrics, 2019). Hence, the authors should better argument their statements and/or highlight the possible differences, if any, between DCI and MYO as for cardio-metabolic consequences.

This statement has been further revised (revised pdf manuscript page 8, lines 431-438).

  • Line 619: the authors should underline that progestins, such as drospirenone and dienogest, appear to mediate the most significant “anti-androgenic” rather “metabolic” clinical effects on symptomatic PCOS women owing to their pharmacological characteristics.

This statement has been edited as suggested (revised pdf manuscript page 12, lines 764-766).

  • Line 632: in my opinion spironolactone should be deleted, since its synergistic role in PCOS treatment maybe argued.

This statement on spironolactone has been deleted as recommended (revised pdf manuscript page 13, line 803).

Round 3

Reviewer 2 Report

in my opinion, the manuscript has been improved thus, it could be suitable for publication. 

Sincerely

Author Response

We thank the Reviewer for helpful comments and suggestions.